# Benefits of Early Integrated and Vocational Rehabilitation in Breast Cancer on Work Ability, Sick Leave Duration, and Disability Rates

**DOI:** 10.3390/healthcare12232433

**Published:** 2024-12-03

**Authors:** Nina Kovacevic, Tina Žagar, Vesna Homar, Bojan Pelhan, Marko Sremec, Tina Rozman, Nikola Besic

**Affiliations:** 1Department of Gynecological Oncology, Institute of Oncology Ljubljana, 1000 Ljubljana, Slovenia; 2Faculty of Medicine, University of Ljubljana, 1000 Ljubljana, Slovenia; vesna.homar@gmail.com (V.H.); nbesic@onko-i.si (N.B.); 3Faculty of Health Care Angela Boskin, 4270 Jesenice, Slovenia; 4Department of Epidemiology and Cancer Registry, Institute of Oncology Ljubljana, 1000 Ljubljana, Slovenia; tzagar@onko-i.si; 5Community Health Centre Vrhnika, 1360 Vrhnika, Slovenia; 6Centre for Vocational Rehabilitation, University Rehabilitation Institute Republic of Slovenia, 1000 Ljubljana, Slovenia; bojan.pelhan@t-2.net (B.P.); marko.sremec@ir-rs.si (M.S.); tina.rozman@ir-rs.si (T.R.); 7Department of Surgical Oncology, Institute of Oncology, 1000 Ljubljana, Slovenia

**Keywords:** breast cancer, integrated rehabilitation, vocational rehabilitation, quality of life, sick leave

## Abstract

Objectives: Vocational rehabilitation plays a vital role in helping breast cancer survivors overcome physical, psychological, and occupational challenges, enabling a smoother return to work and improving quality of life. The aim of this study was to evaluate the effect of vocational rehabilitation as part of early integrated rehabilitation compared to conventional rehabilitation on sick leave duration, work ability, and disability rates. Methods: The study was designed as a prospective, interventional study. We enrolled 435 breast cancer patients, 211 patients in the control group, and 224 in the intervention group. The control group received the conventional rehabilitation as offered to breast cancer patients before the pilot study on individualized, integrated rehabilitation, while patients in the intervention group were referred for additional treatments and vocational rehabilitation. Results: There were no differences between the control and the intervention group of patients in terms of patient demographics, tumor size, disease stage, or oncologic treatment. However, compared to the control group, the intervention group had 50 days shorter sick leave (*p* = 0.002), better work ability (*p* < 0.001), and a lower proportion of patients with disabilities (*p* < 0.001) and better work ability (*p* < 0.001) one year after the beginning of cancer treatment. Vocational rehabilitation was likely associated with shorter sick leave (*p* < 0.069). Conclusions: Integrated rehabilitation was associated with shorter sick leave, and vocational rehabilitation was likely associated with shorter sick leave. Integrated rehabilitation was associated with improved work ability and disability rate.

## 1. Introduction

Cancer and cancer treatment have a major impact on the lives of breast cancer patients [1], as do many other factors that are a consequence of a cancer diagnosis [2]. Breast cancer survivors face a wide range of challenges, including physical, emotional, cognitive, social, and financial difficulties, that significantly impact their quality of life and long-term well-being [3].

Physical consequences include lymphedema, chronic pain, and limited shoulder movement. Changes in appearance, such as breast loss or scarring, can impact body image and self-esteem. Hormonal therapies may induce early menopause, causing hot flashes, mood swings, and vaginal dryness. Cardiovascular and bone health risks also increase. Emotional and psychological impacts may be significant [4]. Fear of recurrence is a pervasive concern among survivors, often leading to anxiety and depression [5]. Cognitive effects include symptoms like memory loss, reduced concentration, and diminished mental clarity following chemotherapy. These effects can interfere with daily functioning and professional tasks [6]. Social and interpersonal impacts are evident in relationship dynamics. The strain of cancer treatment can affect interactions with partners, family, and friends. Fertility issues, a consequence of treatments, affect family planning decisions, particularly for younger survivors [7]. Time off work often leads to lost income, further compounding financial instability. Lifestyle adjustments are necessary for many survivors. Adopting healthier habits, such as improved diet and exercise, is frequently recommended but may be hindered by fatigue or physical limitations [8].

However, for many survivors, the journey to recovery extends well beyond medical treatment. Survivors often face a range of above-mentioned challenges that complicate their return to the workforce and impact their ability to fully reintegrate into their professional lives. Employment is not only a source of financial stability for many individuals but also a critical component of social identity, self-worth, and well-being [9]. The inability to return to work or difficulties in sustaining employment can, therefore, have a profound impact on a survivor’s quality of life [10]. Returning to work as soon as possible is of great benefit to both patients and society. Integrated rehabilitation can restore health and should include medical, psychological, vocational and social rehabilitation [11]. To establish this in Slovenia, we conducted a prospective study with breast cancer patients, which was designed as a pilot study in order to ensure integrated rehabilitation for all breast cancer patients in the future. The results of our early integrated rehabilitation showed that the quality of life improved, the side effects of treatment decreased, and the patient’s needs (problems with global health, physical, emotional, cognitive, and social function, fatigue, and pain) were reduced compared to usual rehabilitation one year after the beginning of cancer treatment [12].

The patient’s occupational activity has a positive influence on her recovery, her psychological and physical well-being, her social security, and her quality of life [13]. Vocational rehabilitation is an organized, professional process, a method of helping people with health problems to return to work and stay employed [13]. There is very little research that has looked at the impact of vocational rehabilitation and return-to-work interventions on breast cancer survivors [14]. The aim of this study was to evaluate the effect of vocational rehabilitation as part of early integrated rehabilitation compared to conventional rehabilitation on sick leave duration, work ability, and disability rates.

## 2. Materials and Methods

### 2.1. Patients

A total of 600 consecutive female breast cancer patients treated at the Institute of Oncology Ljubljana, Slovenia, were included in our prospective interventional pilot study. We enrolled 299 patients in the intervention group with integrated, individualized rehabilitation and 301 patients in the control group with conventional rehabilitation. Inclusion criteria were patients with histologically confirmed invasive breast cancer, all stages, aged between 25 and 65 years, treated at the Institute of Oncology Ljubljana in the years from 2019 to 2022 [12]. Patients were consequently assigned to the control group from December 2019 to March 2021 and to the intervention group from September 2020 to January 2022. While we were enrolling patients in the control group, the institute’s clinical guidelines and integrated rehabilitation pathway were prepared.

Our integrated rehabilitation approach has already been published by Auprih et al. and Cencelj et al. [12,15]. Briefly, all patients filled out 3 standardized questionnaires (European Organisation for Research and Treatment of Cancer) (EORTC) QLQ- C30 (Core 30), EORTC QLQ—BR23 (Breast Cancer Module 23), and National Comprehensive Cancer Network (NCCN) before and half a year and 1 year after the beginning of treatment. The following problems of the patients were monitored: depression, anxiety, cognitive functions, fatigue, insomnia, lymphedema, problems with the shoulder joint, scarring, heart function, lack of female hormones, gynecological problems, sexual problems, muscle and joint pain, smoking, alcohol consumption, diet, pain issues, alopecia, and returning back to work. During a regular check-up with the oncologist, each patient filled in the three standardized questionnaires. Patients from the intervention group had integrated rehabilitation according to their needs, while patients from the control group had standard rehabilitation that was performed in Slovenia before our prospective intervention study.

The socio-economic status was self-reported by the participants in a questionnaire that they received before the operation at the Institute of Oncology Ljubljana. Possible answers were low class, middle class, and higher class. We considered this a good assessment of how (self) perceived (subjective) socio-economic status affects an individual’s life choices and ability to manage their health (at all levels, from prevention to regular visits to the doctor to rehabilitation after treatment). This is based on Townsend’s theorization of relative deprivation, which defines disadvantaged individuals as those who are unable to meet the needs identified by the majority of people in the society in which they live [16].

Those who did not want to participate in the study or could not complete the questionnaire because they did not understand the questions (language, cultural, or cognitive barrier) were excluded. In the present study, data were analyzed for a subgroup of all employed patients: 224 in the intervention group and 211 in the control group.

This study was reviewed and approved by the Protocol Review Board (ERID-KSOPKR-0086/2019) and by the Ethics Committee of the Institute of Oncology Ljubljana (ERIDEK-0102/2019). The study was conducted in accordance with the ethical standards set out in the relevant version of the 1964 Declaration of Helsinki and was conducted with the understanding and consent of all subjects involved. All women signed informed consent.

### 2.2. Rehabilitation

In Slovenia, rehabilitation regularly begins after the end of oncological treatment. The majority of breast cancer patients are referred for spa rehabilitation. Interventions are only carried out if the doctor has identified the patient’s needs. Patients with obvious psychological problems were referred to psychological interventions, while those with lymphoedema were referred to physiotherapy. Side effects were treated when they occurred and were recognized. However, oncologists often overestimate patients’ functional status and underestimate their symptoms [17], so many patients’ needs are often not addressed.

On the other hand, the early integrative rehabilitation performed in our pilot project was more active and based on patients’ needs reported in questionnaires during the course of treatment and immediately followed up by a rehabilitation coordinator, as described elsewhere [12,15]. Integrated individualized rehabilitation began before and during oncological treatment and lasted for one year, tailored to the patient’s needs. Our goal was also to encourage lifestyle change, promoting physical activity and a healthy diet. Guidelines and clinical pathways were based on the NCCN guidelines [4,18,19]. Additional attention was paid to the avoidance of side effects of oncologic treatment. The integrative rehabilitation coordinator counseled patients in the intervention group individually on how to prevent side effects and informed them about the recommended interventions of the multidisciplinary team to help them cope with unmet needs and medical, psychological, vocational, and/or social problems. Integrative rehabilitation started as early as possible, at the beginning or during cancer treatment, with interventions that the patient needed [12]. Patients in the control group also completed all three standardized questionnaires but were not advised by the integrative rehabilitation coordinator on how to prevent side effects. Furthermore, their documentation was not reviewed by a multidisciplinary team, and appropriate interventions were not suggested for their unmet needs.

The documentation of each patient in the intervention group was reviewed three times by the multidisciplinary team: before the start of oncologic therapy, after 6 months, and after 12 months. Depending on the patient’s needs, the team proposed appropriate interventions, about which the patients were informed by the coordinator. The interventions were carried out at the Institute or by other providers. At the institute, patients received physiotherapy, psychotherapy, clinical nutrition, a gynecologist, a pain clinic, acupuncture, and online sports exercises guided by a kinesiologist twice a week, an online yoga practice once a week, and online workshops on healthy diet. Patients were also advised to see their general practitioner (GP). The GP could refer the patient to the health promotion centers, which organize numerous targeted activities and workshops to help patients manage their problems and lead a healthy lifestyle. For employed patients, the third and very important pillar of support was the team of the Centre for Vocational Rehabilitation (CVR) of the University Rehabilitation Institute of the Republic of Slovenia. Of course, all interventions in the intervention group and in the control group were covered by health insurance.

### 2.3. Vocational Rehabilitation

The employed patients from the intervention group were recommended to participate in a vocational rehabilitation program conducted at the CVR. The CVR team includes an occupational medicine specialist, a psychologist, a social worker, and an occupational therapist. Timely referral is very helpful in vocational rehabilitation, so patients are referred to vocational rehabilitation as soon as possible. Patients’ motivation and positive attitude towards return-to-work programs were inclusion criteria for vocational rehabilitation. Contraindications for vocational rehabilitation programs were the same as are routinely used for inclusion in vocational rehabilitation in CVR) for all patients with injuries or other illnesses: lack of motivation, unemployment, poor prognosis, stressful oncological treatment, very poor functioning, dependence on psychoactive substances and/or acute mental illness. In order to achieve successful vocational rehabilitation, the team sought the active involvement of rehabilitants, therapists, employers, the patient’s family, and co-workers.

After admission to treatment, an additional in-depth assessment was carried out, and a specific vocational rehabilitation treatment plan was prepared. Patients had outpatient treatment that involved various members of the vocational rehabilitation team. The treatment was carried out on an outpatient basis five days a week for two weeks. During this time, the rehabilitation team prepared a final report and a rehabilitation plan for return to work. In most cases, extended vocational rehabilitation was required. This means that they required specialized vocational skills training in CVR cabinets before they could return to work or that they received outpatient, long-term support from the CVR team while they worked. From the start, most of the patients returned to work in shorter working hours, combined with sick leave. The CVR team helped patients improve their cognitive function and/or motor activities. The members of the vocational rehabilitation team also helped with insurance claims procedures and organizing life with a disability at home and in the social environment. They also offered them support in adopting a healthy lifestyle: exercise, nutrition, rest, creativity, and social contact. Long-term support needed to return to work was provided by a family medicine specialist, a vocational rehabilitation team, and an oncologist.

The data for the analysis of the duration of illness were from the Slovenian Pension and Disability Insurance Institute, and the data on work ability and disability rate were from the National Institute of Public Health.

### 2.4. Statistical Analysis

Chi-square and ANOVA tests and linear and multivariate regression analyses were used. The dependent variable (outcome) was the duration of sick leave or the ability to work. Regression analysis was performed first for each individual variable and then for the same variable and group (control/intervention) simultaneously. All statistical analyses were performed using version 27 of the statistical software SPSS and Software R version 4.2.2. *p*-values below 0.05 were considered statistically significant.

## 3. Results

### 3.1. Patient Population

Table 1 shows that there were no statistical differences between the intervention and control groups in terms of patient demographics, tumor size, disease stage, or oncologic treatment. There were no significant differences between the intervention and control groups of patients in terms of tumor stage or neoadjuvant therapy (slightly higher proportion in the control group) or chemotherapy (slightly higher proportion in the intervention group). Therefore, both groups have similarly ill patients.

The only exception was the area of residence. At the time the study was planned (before the COVID-19 pandemic), it was assumed that patients would come to the gym to exercise under the guidance of a kinesiologist. Therefore, only patients from urban and suburban areas were included in the intervention group, as they would otherwise take too much time to get to the gym. A statistical comparison regarding all other factors listed in Table 1 showed no statistically significant differences between the urban or suburban group and the rural group of patients.

### 3.2. Vocational Rehabilitation

Out of 224 employed patients in the intervention group, 195 patients were referred to the CVR, as 16 patients refused vocational rehabilitation and 13 had a contraindication for it. 160/195 (82%) patients were examined in the CVR’s outpatient clinic for vocational rehabilitation. In the CVR, 63 patients were examined at the beginning of oncological treatment, 80 patients six months after the beginning of treatment, and 17 patients one year after the start of oncological treatment. A total of 35 patients were admitted but did not participate in the vocational rehabilitation program.

None of the patients from the control group were referred to the CVR.

### 3.3. Sick Leave

The return-to-work rate for all enrolled patients (N = 435) was 55% one year after the beginning of oncologic treatment. For the patients in the intervention group, it was 57% and 52% for the patients in the control group (*p* = 0.17).

The median sick leave of all patients who participated in vocational rehabilitation at the beginning of oncological treatment, after 6 months, and after 12 months was 266, 386, and 351 days, respectively. For patients who refused vocational rehabilitation or had a contraindication for it, the average sick leave was 157.5 days. The median sick leave of patients who were admitted but did not participate in vocational rehabilitation (N = 35) was 331 days. The median sick leave of all patients from the intervention and control group who did not receive vocational rehabilitation (N = 253) was 345 days.

The sick leave in the intervention group was 50 days shorter than in the control group (*p* = 0.002). On average, patients in the intervention group were on sick leave for slightly longer than 290 days, and those in the control group for slightly longer than 340 days. In a subgroup of patients who received chemotherapy, sick leave was 43 days shorter in the intervention group than in the control group (*p* = 0.03) (Figure 1).

### 3.4. Univariate Logistic Regressions

The duration of sick leave was statistically associated with patient group (control/intervention), patient education, breast surgery, lymph node surgery, breast reconstruction, and chemotherapy (Table 2).

The duration of sick leave was not statistically associated with vocational rehabilitation, patient age, socio-economic status, with whom they live, disease stage, concomitant diseases, tumor size, neoadjuvant systemic therapy, hormonal therapy, anti-HER-2 therapy, and external beam radiotherapy.

### 3.5. Multivariate Logistic Regressions

The duration of sick leave was statistically associated with patient group (control/intervention), patient education, breast reconstruction, and chemotherapy (Table 3).

A statistical trend was found for a correlation between shorter sick leave and vocational rehabilitation and a correlation between longer sick leave and hormone therapy. The duration of sick leave was not statistically related to patient age, socio-economic status, with whom they live, stage of disease, concomitant diseases, tumor size, neoadjuvant systemic therapy, breast surgery, lymph node surgery, anti-HER-2 therapy, and external beam radiotherapy.

### 3.6. Working Ability and Disability Rate

Patients in the intervention group had a better working ability and a lower disability rate compared to the control group (Table 4).

Patient group (intervention or control) and socio-economic status were associated with work ability and disability rate (*p* < 0.05). Patients with a master’s degree or PhD had better work ability compared to less educated patients (*p* < 0.01).

## 4. Discussion

Breast cancer survivors have a lower work capacity compared to the healthy population [14]. Islam et al. reported in a systematic literature review that return to work was correlated with sociodemographic factors, disease-related factors, treatment factors, psychological factors, and work-related factors [20]. The results of our multivariate regression analysis confirmed their findings. On the other hand, higher education is known to be associated with many different socio-demographic factors. Our patients with higher education had shorter sick leave than those with lower education. Similarly, Sun et al. reported in a literature review that individual patient characteristics and societal and cultural factors, health and well-being, symptoms and function, work demands, and work environment correlate with a return to work [14]. Our more educated patients were likely to be more financially independent and had fewer symptoms that would cause them problems on return to work than the less educated patients. Islam et al. also found that breast cancer survivors were more likely to return to work if they were financially independent [20], as was the case with our better-educated patients.

The duration of sick leave during breast cancer treatment depends, among other things, on how much and how long the economic support for sick leave is granted in different countries. In Norway, Sweden, and Denmark, patients receive 12 months of support, and in the Netherlands, 24 months [21]. In Slovenia, the duration of sick leave is not specified by law, but patients can also receive longer sick leave and financial support if the doctor appointed by the Institute for Health Insurance determines that the patient is entitled to sick leave and financial support equal to 80% of their salary [22]. In a systematic review, Islam et al. reported that the return to work for patients one year after breast cancer diagnosis ranged from 43% to 93%. In our study, return to work was 55% one year after starting oncologic therapy, and there was no statistically significant difference between the intervention and control groups of patients [20]. However, the results of our study show that patients were off sick on average 50 days less after integrated rehabilitation than after conventional rehabilitation. This is not at all surprising, as our approach with multidisciplinary early integrated rehabilitation reduces the side effects of treatment and reduces patients’ needs and also improves quality of life one year after the beginning of cancer treatment compared to conventional rehabilitation [12]. Depending on the patient’s needs, our integrative rehabilitation included medical, psychological, vocational and social rehabilitation. In addition, an experienced nurse, who was our rehabilitation coordinator, took care of the patients and was always available for questions and help. She educated patients about the prevention and treatment of side effects, lifestyle changes and the measures recommended by the multidisciplinary team for integrative rehabilitation.

Various interventions and treatments have been proposed to reduce the duration of sick leave: physiotherapy, two hours of daily physical activity, running, individual exercise, sports, nutritional counselling, dietary counselling, psychoeducation, self-management programme, thermal water treatment, support from healthcare professionals in a hospital or in the community, multidisciplinary interventions, and vocational counselling. These interventions lasted between two weeks [23] and 33 weeks [24] and were delivered by physiotherapists, psychologists, nurses, or multidisciplinary teams [11,23,24,25,26,27,28,29,30].

In a systematic review and meta-analysis, Algeo et al. reported a statistically significant difference in the ability to perform work and family activities in only one of nine included studies [28]. Morgues et al. included 181 patients with non-metastatic breast cancer who were less than 9 months post-completion of chemo/radiotherapy and had no contraindications to physical activity or cognitive impairment. The intervention group received spa treatment in combination with dietitian counseling, while the control group received dietitian counseling only. They found that the intervention group had a greater ability to perform occupational and family activities after a two-week multi-component spa program one year after starting treatment [23]. Similarly, our study also showed that patients in the intervention group had shorter sick leave than patients in the control group. This was most likely the result of the individualized multidisciplinary integrated rehabilitation, which improved quality of life, reduced side effects of treatment, and reduced unmet needs of patients compared to the control group [12]. Similarly, the study by Van Weert et al. showed that multicomponent cancer rehabilitation programs had positive effects on quality of life, physical performance, and muscle strength [25].

In contrast to Morgues’ method of rehabilitation after completion of oncological treatment [23], we advocate early integrative rehabilitation because it is better to prevent patients’ problems than to treat them. The recently published study by Sanft et al. also speaks in favor of early integrative rehabilitation, which is already carried out during cancer treatment. Their patients with stage I-III breast cancer were randomly assigned to either usual care or a home-based exercise and nutrition intervention with counseling sessions by oncology-certified dietitians during systemic chemotherapy. Their study showed that the patients who were physically active and had a proper diet during oncology treatment had better outcomes than the control group. Among patients who received neoadjuvant chemotherapy, the probability of pathological complete response was higher in the intervention group than in the control group (53% versus 28%; *p* = 0.037) [31]. This suggests that early multidisciplinary integrative rehabilitation can influence the efficacy of oncologic treatment and the prognosis of breast cancer patients. Our patients also received such rehabilitation treatment during chemotherapy.

In addition to physical activity and nutritional interventions, fatigue and anxiety during cancer treatment have been shown to be reduced by integrative medicine interventions such as mindfulness-based stress reduction, mindfulness-based cognitive therapy, tai chi, qigong, mindfulness-based interventions for anxiety, yoga, hypnosis, relaxation therapies, music therapy, reflexology, acupuncture, and/or lavender essential oils [32]. Our patients had the opportunity to attend an online yoga class once a week, which could contribute to their well-being and psychological resilience.

The question arises as to what role vocational rehabilitation plays in the duration of return to work. Tingulstad et al. reported in a systematic review and meta-analysis of relatively small randomized controlled trials from Northern Europe that vocational return-to-work interventions did not provide conclusive evidence for these return-to-work interventions compared to usual care [21]. A 2015 Cochran analysis reported only five randomized controlled trials that included multidisciplinary interventions combining professional counseling, patient education, patient counseling, behavioral training with biofeedback, and/or exercise. Moderate quality evidence showed that multidisciplinary interventions in 450 patients that included physical, psychoeducational, and/or vocational components were associated with a 1.11 higher return to work rate than usual care [11]. However, a recent Cochrane analysis of non-medical interventions to improve return to work in cancer patients found that these multidisciplinary or physical interventions were likely to improve return to work compared to usual care. The study included six randomized control trials comparing multidisciplinary interventions with usual care. It was shown with moderate certainty that multidisciplinary interventions were associated with a 1.23 higher return-to-work rate compared to usual care in 497 patients. The analysis also showed that physical and multidisciplinary interventions were likely to increase return to work, whereas psychoeducational interventions were likely to make little or no difference in return to work. Furthermore, the evidence for vocational interventions on return to work was very uncertain. De Boer et al. proposed that effective return to work in cancer patients should involve multidisciplinary interventions that include physical, psychoeducational, and vocational components as needed, preferably tailored to the patient’s needs [27]. Similar to the proposal by de Boer et al., the interventions for our intervention group of patients were already based on the patients’ stated needs in the questionnaires and included a physical, psychoeducational, and vocational component and were tailored to the patient’s needs by a multidisciplinary integrative oncology rehabilitation team. We believe that contraindications to vocational rehabilitation should be considered prior to referral. In addition, vocational rehabilitation is not effective if there is a lack of motivation to work, unemployment, a poor prognosis, stressful oncological treatment, very poor functionality, dependence on psychoactive substances, and/or acute mental illness. Our study showed that integrated rehabilitation was associated with shorter sick leave, and vocational rehabilitation was likely associated with shorter sick leave. Integrated rehabilitation was associated with improved work ability and a lower disability rate.

Thus, our study suggests that early integrated multidisciplinary rehabilitation has a major impact on shorter sick leave, but we could not demonstrate that vocational rehabilitation was related to the length of sick leave. There are several possible reasons for this. One of them could be that only 28% of patients received vocational treatment at the beginning of oncology treatment when it is most beneficial. In 45% of patients, the start of vocational rehabilitation was postponed by 6 or 12 months due to the heavy burden of the disease or the demanding treatment. These patients were unable to receive optimal support from the vocational rehabilitation team when this support is moste effective. In addition, some patients who were self-employed or who were not expected to benefit from vocational rehabilitation because they needed to start work as soon as possible did not want to participate in vocational rehabilitation at all. Of course, they were on sick leave for a shorter period than other patients. On the other hand, patients who were reluctant to return to work or who wanted to take advantage of a long period of sick leave did not opt for vocational rehabilitation at all.

Groeneveld reported that patients felt that the support they received from an occupational health specialist was inadequate [29]. This was probably because occupational health specialists, who were experts in occupational aspects, had limited insight into the diagnosis, prognosis, and treatment of cancer patients and the consequences of cancer for work because they had no contact with the treating physicians of their clients [33]. However, in our study, the occupational health specialists were members of a multidisciplinary integrative oncology rehabilitation team and had information about the diagnosis, prognosis, and treatment of cancer patients, enabling them to understand patients’ needs and concerns. We believe that our occupational health specialists, with the help of their team, which included a psychologist, a social worker, and an occupational therapist, were, therefore, able to play an important role in effectively shortening the sick leave of our breast cancer patients. Long-term support is needed when returning to work. In our country, this is provided by a GP and a vocational rehabilitation team.

Our results show that early involvement in vocational rehabilitation shortens sick leave. Help and support from a supervisor and support from colleagues is a very important part of the work environment for breast cancer patients, who often have treatment-related symptoms and functional limitations [34,35]. We also believe that returning to work as soon as possible is very beneficial for patients, also because it is extremely difficult to integrate into the work process after a long absence due to lost skills and contacts with colleagues. Our vocational rehabilitation effectively shortened the sick leave periods of patients who received vocational rehabilitation at the beginning of their oncology treatment, as their team carried out medical, psychological, social, and vocational rehabilitation.

The work ability of our patients in the intervention group was better than in the control group one year after the beginning of oncology treatment, and a lower proportion of patients in the intervention group were unable to work one year after the beginning of treatment than in the control group. However, one year after the start of oncology treatment is too short a period to assess the degree of work ability and disability, as the formal procedures for determining work ability and disability pension are very long in our country. In addition, patients in our country also have the opportunity to receive replacement income through the social security system or retire, but the income of retirees is much lower than the financial protection of sick leave, which may affect our disability rate. On the contrary, the replacement income of survivors in France can encourage retirement [35]. Different social contexts influence return to work [14]. For example, Sweden is a country with one of the highest employment rates for people over 55, which may reflect different social values [14,36].

Our study also has some limitations that need to be acknowledged. It is a single-center study and a non-randomized study, as we included patients from two consecutive time periods in our study (the patients in the first part of the study were all assigned to a control group, and the patients in the second period to an intervention group). The selection of patients for the control and intervention groups occurred in different time periods. A possible bias might have been that therapeutic guidelines and medical practices for oncological treatment changed between both groups of patients during the course of the study. But during the study period, the institute’s oncology guidelines did not change, as evidenced by the fact that there was no statistical difference in the proportion of patients treated with neoadjuvant chemotherapy, conservative surgery, reconstruction, postoperative chemotherapy, anti-HER-2 therapy, irradiation, or hormonal therapy. However, the impact of the COVID-19 pandemic cannot be neglected, as access to all health services (e.g., psychotherapy and health promotion centers) was not always equally good, patients could not receive all the proposed interventions in the same proportion as under normal circumstances.

A possible bias is that, according to our inclusion criteria, significantly more motivated patients were included in vocational rehabilitation. However, there are two groups of patients who were not included: those who were not motivated for the work program, as well as those who were on sick leave for only a very short time because, due to the nature of their work, they could not miss work or because they were self-employed and could not afford sick leave. The former were on sick leave for a whole year, while the latter were on sick for a very short time leave, and the average sick leave for both groups was 157.5 days.

In addition, the living areas of the two patient groups were not identical. Unfortunately, our study was not designed from the beginning to test the effects of vocational rehabilitation on sick leave, as we wanted to introduce integrative rehabilitation for patients with breast cancer in our country, so our study served as a pilot study.

These methodological limitations may have introduced bias into the findings and limited the generalizability of the results. Future research should aim to address these issues by ensuring that patient selection is conducted within the same timeframe and employing more inclusive criteria to improve group comparability.

Despite these limitations, the study provides valuable insights into the potential impact of vocational rehabilitation on breast cancer survivors, but the interpretation of results should be approached with caution.

## 5. Conclusions

Integrated rehabilitation and vocational rehabilitation have a positive impact on reintegration in the work process, the ability to work, and the percentage of disability. Patients in the intervention group had shorter sick leave, better work ability, and a lower percentage of disability one year after the start of cancer treatment compared to the control group. Vocational rehabilitation was likely associated with shorter sick leave. Future research should focus on long-term outcomes and personalized approaches to integrated and vocational rehabilitation, as well as their applicability to diverse patient populations. In practice, the early implementation of interdisciplinary rehabilitation programs, employer collaboration, and training for healthcare providers and patients are essential to maximize the benefits of these interventions.

## Figures and Tables

**Figure 1 healthcare-12-02433-f001:**
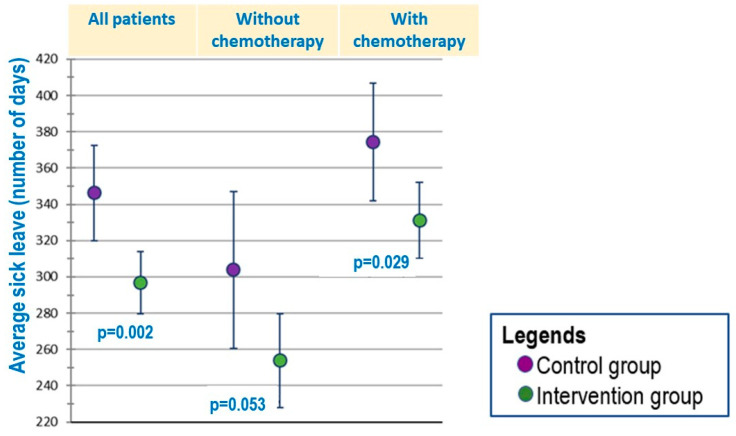
Average sick leave among all patients, patients with chemotherapy and without chemotherapy.

**Table 1 healthcare-12-02433-t001:** Patient demographics, tumor characteristics, and oncologic treatment data.

Factor	Subgroup	Control Group (N = 211)	Intervention Group (N = 224)	*p*-Value *
Mean age (years)		47.9	48.9	0.24
Living areas	Urban	93	126	0.025
	Suburban	36	36	
	Rural	82	62	
Education	Primary school	13	12	0.75
	Secondary school	76	88	
	Graduate	105	103	
	MSc or PhD	17	21	
Socio-economic status	Low	23	21	0.66
	Middle	155	158	
	Higher	33	45	
With whom she lives	Alone	11	23	0.60
	With partner only	34	44	
	Partner and children	118	122	
	With children only	18	17	
	Other	30	18	
Tumor size (mm)		28.6	25.1	0.07
Tumor stage	In situ	4	3	0.18
	I	92	88	
	II	69	94	
	III	37	26	
	IV	9	13	
Concomitant diseases	No	84	98	0.46
	Yes	127	126	
Neoadjuvant chemotherapy and/or anti-HER-2 therapy	No	154	175	0.26
	Yes	57	49	
Breast surgery (N = 423)	Mastectomy	99	99	0.83
	Tumorectomy	106	119	
Breast reconstruction	No	145	151	0.40
	Yes	66	73	
Breast external beam radiotherapy	No	51	50	0.73
	Yes	160	174	
Chemotherapy	No	127	123	0.27
	Yes	84	101	
Anti-HER2 therapy	No	182	191	0.63
	Yes	29	33	
Hormone therapy	No	45	45	0.84
	Yes	166	179	

* The *p*-value is shown for testing the differences between the control and intervention groups (*t*-test for mean age and tumour size and chi-square test for the other variables).

**Table 2 healthcare-12-02433-t002:** Factors associated with duration of sick leave one year after the beginning of oncologic therapy by univariate and multivariate logistic regressions.

	Univariate		Multivariate	
Factor	β	*p*-Value	β	*p*-Value
Intervention group	−51.77	0.001	−81.06	<0.001
Vocational rehabilitation during the first year	−19.69	0.25	−40.21	0.09
Education: Primary school	−25.72	0.552	6.32	0.879
Education: Secondary school	−15.56	0.598	4.51	0.874
Education: High school	−25.62	0.384	−14.53	0.605
Education: Masters Degree or Doctor of Philosophy	−104.69	0.006	−94.61	0.001
Tumor size (each mm)	0.682	0.119	−0.47	0.343
Breast surgery—Tumorectomy	34.79	0.033	−5.68	0.849
Lymph node surgery—Lymphadenectomy	41.95	0.024	−4.41	0.844
Breast reconstruction—Yes	45.13	0.009	59.24	0.026
Chemotherapy—Yes	70.93	<0.001	72.47	<0.001
Hormonal therapy—Yes	20.25	0.312	33.48	0.096
Radiotherapy—Yes	9.78	0.622	37.23	0.156

**Table 3 healthcare-12-02433-t003:** Multivariate logistic regression of vocational rehabilitation and independent factors for the duration of sick leave.

Factor	β	*p*-Value
Intervention group	−77.96	<0.001
Vocational rehabilitation during the first year	−42.40	0.069
Education: Masters Degree or Doctor of Philosophy	−86.03	0.002
Breast reconstruction—Yes	46.18	0.006
Chemotherapy—Yes	63.20	<0.001

**Table 4 healthcare-12-02433-t004:** Working ability of employed patients.

Working Ability	No Disability	With Disability	*p*-Value
Control group	159 (75.3%)	52 (24.7%)	<0.001
Intervention group	210 (93.7%)	14 (6.3%)	

## Data Availability

All data needed to replicate our analyses are available upon request from the corresponding author.

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
