# Peer review of "Benefits of Early Integrated and Vocational Rehabilitation in Breast Cancer on Work Ability, Sick Leave Duration, and Disability Rates"

_healthcare, 2024, doi:10.3390/healthcare12232433_

Round 1
Reviewer 1 Report
Comments and Suggestions for Authors
Dear Authors
Thanks for selecting this important issue for study.
1-It is better to explain more exclusively the consequences of breast Cancer in survivors.
2-How did you select patients for inclusion in each group?
3-How long did the intervention take in the case group? Please clear it in the method part.
4-Did you get informed consent from patientsØŸ
5-Since your inclusion criteria are the patients' motivation and positive attitude towards the return to work program, you essentially include more motivated patients in the group, causing bias.
5-How did you classify the socio-economic status of the participants?
6-What does the abbreviation CVR refer to?
Author Response
Thank you for your thoughtful and constructive comments on our manuscript. We greatly appreciate your recognition of the study's importance. Your insights have provided valuable feedback for improving the clarity and rigor of our work. Alongside the reviewers' comments, we have provided a detailed, point-by-point explanation of the revisions made to the manuscript in the attached Word document .

Reviewer 2 Report
Comments and Suggestions for Authors
The manuscript presented for review is an interesting analysis of the employment of vocational rehabilitation on breast cancer patients. The idea is interesting and the presentation is well organized.
My concerns about this study however refer to the way the parient were selected- the patients in the 2 groups are selected in different periods of time which may not have the same overall conditions of treatment and rehabilitation. I have to point out that the therapeutic guidelines change quite frequently and guidelines separated by a few years may not be the same
Also some of the exclusion criteria for intervention group are demanding chemotherapy and poor prognosis. Both of these suggest that in the control group we have sicker patients which by definition may require longer sick leave - this raises the question about how relevant are the results and what exactly is the connection between vocational rehabilitation and duration of sick leave.
Unfortunately both of these are methodological errors and can not be corrected at this time. But after accepting these limitations and perhaps limiting the overevaluation of the impact of this intervention the article may be published after minor revision.
Author Response

(The authors gave the same response as above.)

Reviewer 3 Report
Comments and Suggestions for Authors
interesting paper but due to the fact that you are bringing so much background
it is difficult to capture the relevant and specific intervention
the terminology is changing I guess as the evolution of the intervention
but I suggest to clarify at the beginning of the manuscript what exactly fits with the title of the paper
my comments are about the difficulty to follow the process in the introduction which impacts the clarity for the methods.

Author Response

(The authors gave the same response as above.)

Reviewer 4 Report
Comments and Suggestions for Authors
I congratulate the authors of the manuscript "Benefits of early integrated and vocational rehabilitation in 435 breast cancer patients shown in a prospective study" for their research. The study makes a significant contribution to the field of oncological rehabilitation. However, I have the following considerations:
-
Title
Instead of emphasizing the number of participants and the type of study in the title, it would be more engaging to highlight the specific areas where these benefits were evaluated. -
Abstract
The abstract could be written in a more compelling and informative manner. The description of the methodology should be aligned with the type of study conducted (e.g., prospective, cohort, case-control, intervention/experimental study). In the results section, it is unnecessary to mention the type of analysis performed; focus instead on the values obtained and how they address the study objectives. -
Objectives
The way the objectives are written, interwoven with the hypothesis, is problematic as it presumes that vocational rehabilitation shortens medical leave, increases work capacity, and reduces disability rates compared to conventional rehabilitation. Scientific research should aim to test the null hypothesis. Therefore, I recommend rewriting the objectives to evaluate the effect of vocational rehabilitation compared to conventional rehabilitation on medical leave duration, work capacity, and disability rates. -
Methodology
The methodological design is unclear. Based on the objectives, the appropriate design would likely be a case-control study, a non-randomized clinical trial, or an interventional study involving participants in vocational or conventional rehabilitation programs. Describing the study merely as "prospective" is insufficient.- The abstract mentions that participants were followed up for at least 12 months. What was the exact follow-up duration defined in the study? (before the start of oncologic therapy, after 6 months and after 12 months -line 101) and more???)?? 101
and after 12 months. - The methodology section needs to be rewritten to ensure greater reproducibility. While the theoretical framework is adequate, the description of actions taken and their sequence lacks clarity.
- The article appears to be structured as a comparison between an intervention and a control group. The method description should be adjusted accordingly.
- The abstract mentions that participants were followed up for at least 12 months. What was the exact follow-up duration defined in the study? (before the start of oncologic therapy, after 6 months and after 12 months -line 101) and more???)?? 101
-
Results
The section includes the following confusing statement:
"3.2. Vocational rehabilitation: Of 211 working patients, 195 patients were referred to the CVR, as 16 patients refused vocational rehabilitation or had a contraindication for it."
Does the 211 patients refer to the control group? Were they included in the intervention group? I suggest standardizing the description of the groups—either by type of rehabilitation or by explicitly stating in the methodology which group represents the intervention and which represents the control.The results section is unclear and needs revision for better clarity.
-
Discussion
It is unnecessary to repeat the findings of the current study in the discussion section. Instead, focus on positioning these findings within the context of existing literature. This would make the text more concise and fluid.
Author Response

(The authors gave the same response as above.)

Round 2
Reviewer 3 Report
Comments and Suggestions for Authors
thanks for your thorough review the article
is more strong with more relevant information and connection